# T-Cell Immune Responses to SARS-CoV-2 Infection and Vaccination

**DOI:** 10.3390/vaccines12101126

**Published:** 2024-09-30

**Authors:** Samuele Notarbartolo

**Affiliations:** Infectious Diseases Unit, Fondazione IRCCS Ca’ Granda Ospedale Maggiore Policlinico, 20122 Milan, Italy; samuele.notarbartolo@policlinico.mi.it

**Keywords:** SARS-CoV-2 infection, COVID-19 vaccine, human T cells, immunological memory, antigen-specific T cells, cross-reactive T cells, hybrid immunity, resident memory T cells, vaccine improvement

## Abstract

The innate and adaptive immune systems collaborate to detect SARS-CoV-2 infection, minimize the viral spread, and kill infected cells, ultimately leading to the resolution of the infection. The adaptive immune system develops a memory of previous encounters with the virus, providing enhanced responses when rechallenged by the same pathogen. Such immunological memory is the basis of vaccine function. Here, we review the current knowledge on the immune response to SARS-CoV-2 infection and vaccination, focusing on the pivotal role of T cells in establishing protective immunity against the virus. After providing an overview of the immune response to SARS-CoV-2 infection, we describe the main features of SARS-CoV-2-specific CD4^+^ and CD8^+^ T cells, including cross-reactive T cells, generated in patients with different degrees of COVID-19 severity, and of Spike-specific CD4^+^ and CD8^+^ T cells induced by vaccines. Finally, we discuss T-cell responses to SARS-CoV-2 variants and hybrid immunity and conclude by highlighting possible strategies to improve the efficacy of COVID-19 vaccination.

## 1. Introduction

The T cell-mediated immunity is key to controlling intracellular pathogens, such as viruses. CD8^+^ T cells directly kill infected cells, whereas CD4^+^ T cells provide “help” through cytokine production and optimize durable and effective CD8^+^ T-cell and humoral responses [1].

Naïve T cells are activated in secondary lymphoid organs upon interacting with antigen-loaded dendritic cells (DCs). DCs present antigens to T cells by loading short peptides onto the major histocompatibility complex (MHC). They stimulate the specific T-cell receptor (TCR), provide costimulatory signals, and produce polarizing cytokines and metabolites that regulate the function and migration of activated T cells. When stimulated, activated naïve T cells start to proliferate and differentiate into effector T cells that enter the circulation and travel to peripheral tissues, where they carry out their protective function [2]. After clearance of the pathogen, most effector T cells die by apoptosis, and only a small fraction of them persist as central memory, effector memory, and resident memory T cells that will provide an enhanced systemic or local immune protection to the host when reexposed to the same antigen [3]. The immunological memory is a peculiar property of the adaptive immune system and underlies vaccine-induced protective immunity.

Here, we aim to review the current knowledge of the immune response to SARS-CoV-2 infection and vaccination, revolving around T cells and cellular immunity. First, we will give an overview of the immune response to SARS-CoV-2 infection, including a brief description of the role of innate immunity in controlling the virus. We will then focus on the role of T cells, highlighting their capacity to provide immune protection in the absence of humoral immune responses. We will describe the different features of SARS-CoV-2-specific CD4^+^ and CD8^+^ T cells generated in patients with mild and severe COVID-19 and compare them with Spike-specific CD4^+^ and CD8^+^ T cells induced by vaccination. Finally, we will highlight some open questions and future perspectives in the field, discussing possible strategies to improve the efficacy of COVID-19 vaccines.

We are aware and would like to remind the readers that the large majority of published studies on the topic have been conducted on the peripheral blood of patients with COVID-19 and vaccinated individuals, which may convey a partial view of the immune response to SARS-CoV-2 infection and vaccination since most of immune cells are not in circulation [4]. Nonetheless, by being relatively easy to perform and minimally invasive, investigating the immune responses in peripheral blood has allowed longitudinal sampling to monitor the evolution of the effector response and the generation of immunological memory against SARS-CoV-2. Moreover, by being performed on large cohorts of individuals, it has enabled the scientific community to identify correlates of protective immunity, generating fundamental knowledge that has been instrumental in tackling the COVID-19 pandemic.

## 2. Overview of the Immune Response to SARS-CoV-2 Infection

The severe acute respiratory syndrome coronavirus 2 (SARS-CoV-2) is a single-stranded positive-sense RNA virus that causes the Coronavirus Disease 2019 (COVID-19). It belongs to the *Sarbecovirus* subgroup of the *Betacoronavirus* genus, which also includes the SARS-CoV-1 and MERS viruses that have caused epidemic and pandemic outbreaks of diseases in the last 20 years [5].

SARS-CoV-2 infection occurs without symptoms or with mild ones in most individuals and is usually resolved in 10–20 days [6]. When symptomatic, it results in an influenza-like illness that can eventually progress to interstitial pneumonia, acute respiratory distress syndrome, and death. The most common COVID-19 symptoms include fever, dry cough, and shortness of breath, followed by fatigue, myalgias, headache, rhinorrhea, anosmia, ageusia, diarrhea, nausea, and vomiting [7,8,9].

The optimal immune response to SARS-CoV-2 requires innate and adaptive immunity to function coordinately (Figure 1). The proper setup of an innate immune response mediated by type I and type III interferons (IFNs) is critical for establishing an effective antiviral adaptive immune response [10,11]. The innate immune system can detect the presence of SARS-CoV-2 and other RNA viruses through Toll-like receptors (TLR3 and TLR7/8) and RIG-I-like receptors (RIG-I and MDA5) [12]. The signaling downstream of these two classes of pattern recognition receptors converges into the activation of IRF3 and IRF7 transcription factors that cooperate with the nuclear factor κB (NF-κB) to produce type I and type-III IFNs. Type I and type III IFNs bind their cognate receptors IFNAR1/IFNAR2 and IFNRL1/IL10R2 on infected cells and antigen-presenting cells (APCs) and initiate a signaling cascade that induces the STAT1/STAT2/IRF9-driven activation of several interferon-stimulated genes (ISGs), which interfere with the viral life cycle [13], and proinflammatory cytokines production. In addition, type I IFN can directly act on T cells to support clonal expansion and memory formation in response to viral infections [14,15].

SARS-CoV-2 can inhibit or delay type I/III IFN-mediated immune response by antagonizing type I/III IFN production and the downstream signaling through different mechanisms mediated, sometimes redundantly, by several structural and nonstructural viral proteins [16,17]. Therefore, temporally delayed and lower levels of type I or type-III IFNs are detected in the lungs or the peripheral blood of patients with severe COVID-19 compared with other respiratory infections [18,19]. The suppression of the type I/III response results in higher viral replication, delayed activation of the adaptive immune system, excessive inflammation, and tissue damage [20], which underlie the systemic inflammatory syndrome that characterizes severe COVID-19 cases [21]. Notably, genetic mutations and autoantibodies that interfere with IFN pathways have been detected, respectively, in about 3–5% of younger adults and 15–20% of patients over 70 years old with critical COVID-19 pneumonia [22,23,24], indicating that the IFN-mediated immune response is impaired both by virus-dependent and host-dependent mechanisms in a sizeable proportion of patients with severe disease.

Despite the important role of innate immunity, the best control of SARS-CoV-2 infection and protection from severe COVID-19 is achieved by a coordinated adaptive immune response made by virus-specific neutralizing antibodies (nAbs) and CD4^+^ and CD8^+^ T cells [25]. Indeed, the early induction of SARS-CoV-2-specific T cells and nAbs positively correlates with a better clinical outcome in patients with COVID-19 [26,27] (Figure 1). Antibodies recognize viral epitopes in their native conformation and can block the binding of the viral Spike protein to the ACE2 receptor on human cells, thus preventing infection. Moreover, antibodies can promote the killing of virus-infected cells, either by binding to Fc receptors on NK cells and inducing the antibody-dependent cellular cytotoxicity [28] or by activating the classical pathway of the complement, which leads to the lysis of the infected cells and facilitates their clearance by phagocytes [29]. Antibodies act on the extracellular virus and are most effective when present before the beginning of an infection. On the contrary, T cells cannot recognize a virus until cells are infected, as they need viral epitopes to be processed and presented by MHC molecules on APCs. However, T cells can recognize epitopes derived from any viral protein, thus broadening the repertoire of targetable viral structures [30]. CD8^+^ T cells identify and directly kill infected cells and are critical in eliminating the virus in many viral infections [31]. CD4^+^ T cells contribute to antiviral immunity by at least three distinct mechanisms involving follicular helper T (T_FH_) cells, T_H_1 cells, and cytotoxic (CD4^+^-CTL) cells. T_FH_ cells help the humoral response to sustain the affinity maturation of B cells and the generation of durable antibody responses [32]. T_H_1 cells produce IFN-γ and other cytokines that activate cell-intrinsic antiviral responses in infected cells and phagocytes and promote the recruitment of effector cells at the site of infection. They have been shown to contribute to immune protection against influenza [33] and SARS-CoV-1 [34] infections. Finally, CD4^+^-CTL cells bear a cytotoxic activity similar to CD8^+^ T cells and can directly kill MHC-II-expressing infected cells. They have been detected in several viral infections and associated with protection in patients infected by influenza [33] and Dengue [35] viruses. Notably, MHC-II-expression has been widely detected in the inflamed lung epithelial and endothelial cells of patients with lethal COVID-19 [36], suggesting that CD4^+^-CTL cells may cooperate with CD8^+^ T cells in killing SARS-CoV-2 infected cells.

Several studies on transplanted patients undergoing immunosuppressive therapy confirmed the critical role of the immune system in controlling viral infection [37,38]. During the first wave of the COVID-19 pandemic, up to 90% of patients receiving immunosuppressive drugs due to solid organ transplantation (SOT) needed hospitalization [39,40], compared with 12–15% of the general population, with crude mortality rates of 20–25%. Despite a significant reduction in the mortality of SOT patients in the subsequent waves of the pandemic due to improved clinical management of the disease, their hospitalization and mortality rates remained significantly higher than the general population. Moreover, they only partially benefitted from introducing COVID-19 vaccines [41,42,43,44]. Interestingly, a study reported that treatment with different immunosuppressive drugs may significantly alter the risk of hospitalization in specific SOT settings [38]. The authors speculated about a possible association between certain immunosuppressive drugs (e.g., mycophenolic acid), the impairment of a particular branch of the immune system (i.e., a cytostatic effect on lymphocytes), and the clinical outcome (i.e., worse outcome) [38]. However, additional investigation is required to functionally support this and similar statements.

Important information about the contribution of the different arms of the immune system in controlling SARS-CoV-2 infection comes from large studies performed on individuals with various types of inborn errors of immunity (IEI). These studies showed that IEI patients typically experience prolonged viral shedding and have higher mortality rates compared to the age-matched general population [45,46,47]. However, most IEI patients (90–95%) can resolve the infection, often with mild or moderate symptoms, highlighting redundancies and compensatory mechanisms in the human immune system for host defense against SARS-CoV-2 [45,46,47]. The largest group of patients with IEI analyzed so far was made by individuals affected by deficiency in antibody production, such as common variable immunodeficiency and hypogammaglobulinemia, indicating that SARS-CoV-2 infection can be controlled by the immune system in the absence of nAbs production, likely by T cells. Similarly, therapeutic depletion of B cells by rituximab in patients with multiple sclerosis is associated with an increased risk of developing a severe disease requiring hospitalization but does not significantly correlate with higher mortality rates [48,49,50].

Investigating the opposite scenario, namely, the ability of B cells and antibodies to control SARS-CoV-2 infection without T cells, is more difficult because a complete lack of T cells is incompatible with life. However, evidence about the important role of T cells in the resolution of SARS-CoV-2 infection emerges from various clinical settings. In patients with Acquired Immune Deficiency Syndrome (AIDS) having an active SARS-CoV-2 infection, CD4^+^ T-cell lymphopenia is associated with poorer outcomes [51,52], even when nAbs are produced. Among patients with hematologic cancer, the ones with defective CD4^+^ and CD8^+^ T-cell responses had the highest mortality, regardless of the presence of B-cell responses [53]. On the contrary, in patients having compromised humoral immunity due to the disease or therapy, the presence of SARS-CoV-2-specific CD8^+^ T cells was associated with improved survival [53]. This is consistent with the observation that CD8^+^ T cells can control viral loads upon rechallenge with SARS-CoV-2 in convalescent rhesus macaques with waning antibody titers [54]. The ability of T cells to control SARS-CoV-2 infection in the absence of humoral response is corroborated by the identification of SARS-CoV-2-specific T cells in the absence of seroconversion in asymptomatic individuals exposed to the virus and in some patients with paucisymptomatic COVID-19 [55,56]. Notably, we and others have reported cases of patients with COVID-19 who failed to mount a T-cell response, measured by either T-cell clonal expansion [57] or antigen-specific stimulation [25], and succumbed to the disease despite the production of SARS-CoV-2-specific antibodies, further supporting the important role for T cells in resolving SARS-CoV-2 infection.

Altogether, these data demonstrate that the best protection against SARS-CoV-2 infection is provided by integrated and coordinated innate and adaptive immune responses and indicate a superior capacity of T cells to control the virus.

## 3. Qualitative and Quantitative Alterations of T-Cell Populations in Patients with COVID-19

One of the main clinical features of patients with COVID-19 is T-cell lymphopenia correlating with the severity of the disease. A transient lymphopenia occurring during an acute infection before the peak in the T-cell response is a characteristic common to many severe viral infections, can be induced by type I IFN signaling, and may be useful to create the space for a robust virus-specific T-cell response [58]. However, in patients with severe COVID-19, the T-cell lymphopenia can persist weeks after infection or symptoms onset [59]. An increased CD4^+^/CD8^+^ T-cell ratio has been reported in patients with severe COVID-19 [60,61], suggesting that SARS-CoV-2 infection might preferentially impair CD8^+^ T cells, especially from the effector memory population [57]. Initial speculation suggested that the stronger reduction in circulating T cells in patients with severe COVID-19 could be due to an increased migration of T cells to the site of infection. However, the observation that the number of T cells in the bronchoalveolar lavage (BAL) fluid was lower in patients with severe disease than those with moderate disease [62] argues against this hypothesis. Presumably, a temporally dysregulated type I IFN response, the inability to mount a virus-specific T-cell response, the increased apoptosis of T cells, or a combination of these factors may better explain the stronger lymphopenia in patients with severe disease.

Defects in type 1 immune responses [60] and skewing toward type 2 immunity [57,63] have been associated with COVID-19 severity, similar to what was observed in fatal SARS-CoV-1 infections [64] and in an experimental model of influenza infection [65], indicating that an inappropriate immune response to the virus may cause delayed viral clearance and disease deterioration.

The impaired ability to mount an effective antiviral T cell-mediated immune response underlies the uncoordinated adaptive immune response in the elderly [25] that, together with a higher rate of comorbidities, renders age the major risk factor in developing severe COVID-19 (Figure 2). In addition to the mentioned alteration in type I IFN signaling due to the production of autoantibodies and a decline in the antigen presentation potential of APCs [66], the capacity of setting up an adaptive immune response to SARS-CoV-2 in older adults is dampened by the age-related decrease in the repertoire of naïve T cells [67], which causes a contraction in the pool of T cells able to react to a new pathogen (Figure 2). The reduction in naïve T cells is stronger among CD8^+^ than CD4^+^ T cells, possibly because CD8^+^ T cells are more susceptible to the homeostatic proliferation-induced differentiation [68] and can be exacerbated by common chronic infections [69], which leads to the expansion of terminally differentiated and exhausted T cells. Both terminally differentiated CD8^+^ T cells and memory-like CD8^+^ T cells differentiated in response to cytokines include innate-like cytotoxic cells that can be activated in the absence of TCR stimulation [70,71]. Early activation of these bystander-activated cytotoxic T cells can cooperate in viral clearance [72], but their prolonged complement-mediated activation can also contribute to the excessive inflammation and tissue damage characterizing severe COVID-19 [73].

Together, these data indicate that prolonged T-cell lymphopenia and maladapted CD4^+^ and CD8^+^ T-cell responses are associated with severe COVID-19 and that these conditions, for different reasons, occur more frequently in the elders who are, indeed, at higher risk of clinical deterioration upon SARS-CoV-2 infection.

## 4. SARS-CoV-2 T-Cell Antigens and Immunodominant Epitopes

CD4^+^ and CD8^+^ T cells activated by SARS-CoV-2 infection recognize a broad range of viral antigens derived from structural and nonstructural proteins. A meta-analysis of 25 studies identified 1434 nonredundant epitopes recognized by T cells, 399 of which have been defined as immunodominant (110 CD4^+^ and 289 CD8^+^ T-cell epitopes) [74]. These data are constantly updated by the scientific community and, as of September 2024, the Immune Epitope Database [75] has cataloged over 3700 records. The most immunodominant CD4^+^ T-cell epitopes show a high HLA-II binding promiscuity, defined as the capacity to bind multiple HLA allelic variants [76]. Immunodominant epitopes mainly derive from the Spike (S), Nucleoprotein (N), and Membrane (M) structural proteins but also from nonstructural proteins such as nsp3, nsp12, and ORF3a [74,76]. CD4^+^ and CD8^+^ T-cell responses to S, N, M, and nsp3 proteins are highly coordinated, meaning that CD4^+^ and CD8^+^ T cells specific to these proteins are simultaneously generated in most individuals [76]. It has been conservatively estimated that each person recognizes on average 19 different CD4^+^ and 17 CD8^+^ T-cell epitopes [76], but none of the identified epitopes elicit T-cell responses in 100% of the donors tested.

Looking at the distribution of the immunodominant epitopes from S and N proteins, it emerged that the epitopes recognized by CD4^+^ and CD8^+^ T cells have peculiar traits. CD4^+^ T-cell epitopes preferentially concentrate in specific regions, while CD8^+^ T-cell epitopes distribute throughout the antigens’ sequence. Immunodominant CD4^+^ T-cell epitopes in the S protein mainly originate from the N-terminal domain of the S1 subunit, the C-terminus (aa 686–816) and the fusion protein region in the S2 subunit, and a small conserved region (aa 346–365) of the receptor binding domain (RBD) [74,77]. Similarly, CD4^+^ T-cell epitopes in the N protein are localized in the N-terminal and the C-terminal domains, with little contribution from the linker region and protein tails. On the contrary, CD4^+^ T-cell immunogenic regions are distributed along the entire M protein [74]. Interestingly, CD4^+^ T-cell immunodominant regions identified in S and N proteins show a limited overlap with the immunodominant linear regions targeted by antibody responses [74], indicating complementarity between the humoral and cellular immunity and corroborating the correlation between a coordinated adaptive immune response and a favorable clinical outcome.

Several studies have identified SARS-CoV-2-reactive T cells in about 50% of unexposed individuals [78,79,80,81]. These T cells cross-react with conserved epitopes from endemic coronaviruses (NL63, OC43, HKU1, and 229E) and have low-avidity T-cell receptors (TCRs) [82,83]. M-specific CD4^+^ T cells and S-specific CD8^+^ T cells can also originate from cross-reactive cytomegalovirus-specific T cells [84], extending the cross-reactivity to other virus types. Despite a possible association between the presence of cross-reactive T cells, prompter immune responses, and better clinical outcomes [85,86,87], their protective role is still debated [88]. Although the activation of preexisting cross-reactive memory T cells has been observed in patients with COVID-19, most SARS-CoV-2-reactive T cells recognize new epitopes [76,77] through a diversified repertoire of high-avidity TCRs [82,89,90].

These data indicate that T-cell responses to SARS-CoV-2 are broad, multiantigenic, and complementary to humoral responses.

## 5. Kinetics of T-Cell Responses to SARS-CoV-2 Infection and Phenotype of SARS-CoV-2-Specific T Cells

As mentioned before, the prompt activation of SARS-CoV-2-specific T cells positively correlates with a favorable clinical outcome for COVID-19 patients [26]. Studies in macaques support the protective role of T cells against SARS-CoV-2 infection [54]. The identification and quantification of antigen-specific T cells are more complex than the measurement of specific antibodies. They can be performed by different immune assays based either on the ex vivo identification through tetramers and multimers staining or on the in vitro stimulation with peptide pools, representing the whole SARS-CoV-2 peptidome or just selected epitopes, followed by monitoring of cytokine production, cell proliferation, or the expression of activation-induced markers (AIMs) on the cell surface. Both methods have advantages and limitations. Detection by tetramers and multimers binding does not require any T-cell stimulation, and the identification of antigen-specific T cells is independent of the upregulation of specific markers or the production of effector molecules. This technique is precise but does not provide information on T-cell functionality (unless cells are somehow stimulated) and usually allows the detection of a limited repertoire of antigen-specific T cells. Also, it requires knowledge of patients’ HLA haplotypes, although this issue may be partially overcome by using complex libraries [91,92]. On the contrary, the in vitro stimulation with peptide pools, or other sources of specific antigens, enables the monitoring of antigen-specific T-cell effector function and is usually more sensitive. However, the high sensitivity may be paralleled by a lower specificity due to the activation of low-affinity T cells that are not relevant for the in vivo response to the virus, and to the detection of bystander-activated T cells if the antigenic stimulation is too long (e.g., >24 h). Moreover, it is possible that recently in vivo-activated T cells, such as effector cells during acute infection, may not properly respond to the in vitro stimulation or that the in vitro stimulation may alter the original phenotype of T cells.

Despite the mentioned limitations, the results produced by many studies describe defined patterns of induction, expansion, and contraction of the cellular immune response during and after SARS-CoV-2 infection (Figure 3). The timescale of the described immune responses may carry some unavoidable variability since the timing of infection cannot be precisely determined in humans and results are commonly reported as “time after symptoms onset” or “time after positive test”. However, such variability should be limited to a few days and affect the punctual definition only of the very early events of the immune response, since the median incubation period of SARS-CoV-2 infection has been estimated to be 5 days and the great majority (>97%) of individuals develop symptoms within 11 days from viral infection [93]. Notably, these epidemiologic data agree with a study performing a controlled SARS-CoV-2 challenge in healthy volunteers [94]. SARS-CoV-2-specific T cells can be detected already 3–5 days after symptoms onset and expand in the following 10–20 days [25,26,95,96]. Delayed and weaker induction of SARS-CoV-2-specific T cells is observed in patients with severe COVID-19 [25,97], associated with a misfiring of the immune response, characterized by exaggerated and persistent activation of some components of the innate immunity and the lack of type 1 adaptive immune response [63].

SARS-CoV-2-specific CD4^+^ T cells are detected in nearly 100% of infected people and mainly produce T_H_1-associated cytokines, such as IFN-γ and TNF-α, although T_H_17-associated cytokines have also been detected [25,95,98,99]. CD4^+^ T cells from asymptomatic patients or patients with mild disease produce higher amounts of IFN-γ and IL-2 than those from patients with severe COVID-19, indicating a higher functionality and proliferative capacity [98,100]. Clonally expanded CD4^+^-CTL cells have been identified in hospitalized patients [101], but a clear correlation between this T-cell subset and immune protection or disease severity is still missing. Circulating SARS-CoV-2-reactive T_FH_ (cT_FH_) cells, specific for the S, N, and M proteins, are also found upon infection and correlate with nAbs production [102,103,104]. Notably, post mortem examination of thoracic lymph nodes and spleen highlighted a strong reduction in BCL-6^+^ germinal center B cells that was associated with an early block of BCL-6^+^ T_FH_ differentiation [105], demonstrating the relevance of T_FH_ cells in supporting the humoral immune response to SARS-CoV-2 infection. Interestingly, the S-specific cT_FH_ cells have been reported to be more abundant than S-specific T_H_1 cells, while the cT_FH_/T_H_1 ratio was inverted for the N-specific CD4^+^ T cells [98], suggesting a specialization of effector responses to different viral antigens.

SARS-CoV-2-specific CD8^+^ T cells are identified in about 70% of infected individuals, and their induction strongly and significantly correlates with a better clinical outcome in patients with COVID-19 [25]. During an acute infection, SARS-CoV-2-specific CD8^+^ T cells mainly display an effector memory phenotype, characterized by the lack of CD45RA and CCR7 expression and producing high amounts of effector molecules, such as granzyme B, perforin, and IFN-γ [98,106,107]. Some studies suggested a possible association between the frequency of exhausted CD8^+^ T cells and COVID-19 severity, but they are not supported by consistent data in the literature [108]. Indeed, T-cell exhaustion normally results from the chronic stimulation of T cells and would not fit well in the context of an acute viral infection. Much confusion derives from the fact that in many papers, researchers tend to refer to T cells expressing inhibitory receptors and other exhaustion-associated molecules as exhausted T cells. However, inhibitory receptors are usually upregulated in recently activated T cells to restrain their effector function on time and avoid excessive immune responses. Instead, T-cell exhaustion is a permanent (unless treated) dysfunctional state requiring epigenetic changes and metabolic reprogramming and marked by an elevated and persistent expression of inhibitory receptors. Nonetheless, patients with mild COVID-19 tend to have a higher frequency of total and SARS-CoV-2-specific CD8^+^ memory precursor effector T cells than those with severe disease [57,106], suggesting an impaired or delayed generation of memory CD8^+^ T cells in patients with severe COVID-19.

Two–three weeks after symptoms onset, the frequency of circulating SARS-CoV-2-specific T cells starts declining [109] (Figure 3). The kinetics of the contraction phase differs, at least in part, between CD4^+^ and CD8^+^ T cells: CD8^+^ T cells are progressively reduced starting about 1 month after infection, while the frequency of CD4^+^ T cells is more stable at least until 2 months [25,110]. The reasons underlying the different kinetics of contraction are still debated. They might result from the higher tendency of CD8^+^ T cells to reside in peripheral tissues than in the circulation and from persisting antigenic depots in dendritic cells within lymph nodes that sustain CD4^+^ T-cell activation and proliferation. Nevertheless, more recent studies analyzing the presence of memory T cells within 8 months from the infection calculated a comparable half-life for CD4^+^ (t_1/2_ 94–207 days) and CD8^+^ (t_1/2_ 125–196 days) T cells [111,112], indicating that the different kinetics are limited to the first part of the contraction phase. Interestingly, a longitudinal study investigating the presence of SARS-CoV-2-specific CD4^+^ T cells at 6 to 15 months after infection calculated a t_1/2_ of 377 days, indicating a flattening of the contraction phase and suggesting the establishment of virus-specific long-term memory T cells.

Data collected in the first months after resolution of SARS-CoV-2 infection indicate that SARS-CoV-2-specific CD4^+^ T cells are mainly central memory (T_CM_) and effector memory (T_EM_) T cells, with a trend of T_CM_ increasing and T_EM_ decreasing over time [111,112]. On the contrary, SARS-CoV-2-specific CD8^+^ T cells are mainly effector memory (T_EM_) and effector memory CD45RA^+^ (T_EMRA_) T cells, with an increasing accumulation of T_EMRA_ over time [107,111,112]. Nonetheless, it has been reported that the majority of SARS-CoV-2-specific CD8^+^ T cells one year after infection express the transcription factor cell factor 1 (TCF-1) but not the thymocyte selection-associated high-mobility group box (TOX) [107], suggesting they are endowed with self-renewal capacity and not terminally differentiated or exhausted. Moreover, a small population of SARS-CoV-2-specific CD4^+^ and CD8^+^ stem-cell memory T cells (T_SCM_) have been detected postinfection up to 10 months after symptoms onset, hinting at the formation of long-lasting virus-specific memory cells [113]. These data are consistent with the observation made in the context of SARS-CoV-1 infection where virus-specific memory T cells are detected over 10 years postinfection [99,114].

Most studies investigated the phenotype and function of circulating SARS-CoV-2-specific T cells from peripheral blood. However, T cells expressing activation markers or showing clonal expansion have been found in the lungs of patients with COVID-19 [62,115,116], suggesting the establishment of T cell-mediated responses in the infected tissues. Although studies investigating the antigen specificity of tissue-infiltrating T cells during the acute infection are still scarce, SARS-CoV-2-specific CD4^+^ and CD8^+^ T cells have been found in the bone marrow, spleen, lymph nodes, lung, and nasal mucosa of COVID-19 patients up to 6 months after infection [117,118,119]. The generation of SARS-CoV-2-specific and cross-reactive resident memory T (T_RM_) cells in the upper and lower airways [117,120] may contribute to the protection against the disease upon reinfection by rapidly recognizing the virus and providing an alarm function also in case of failure or delay of the innate immune response [121,122]. Indeed, a longitudinal study of BAL fluid from 273 patients with severe pneumonia showed an association between the presence of alveolar T cells targeting structural SARS-CoV-2 proteins and a better clinical outcome in unvaccinated patients [123].

These data demonstrate that virus-specific CD4^+^ and CD8^+^ effector T cells are induced in response to SARS-CoV-2 infection and contribute to the resolution of the infection. Infected individuals develop memory T cells that can be detected up to 15 months after infection and are predicted to last for years. SARS-CoV-2-specific memory T cells include circulating and tissue-resident T cells (Figure 4) that can provide multiple layers of enhanced protection against a severe disease upon reinfection.

## 6. SARS-CoV-2-Specific T-Cell Responses to COVID-19 Vaccines

The development of vaccines against SARS-CoV-2 has been the breakthrough for mitigating the severe illness and hospitalization associated with COVID-19. The objective of vaccines is to stimulate an immune response against a pathogen without causing the pathogen-associated disease to train the immune system to face the same pathogen in the context of a natural infection. Four different types of COVID-19 vaccines have been developed and approved. They use as antigen source the inactivated whole virus (e.g., CoronaVac by Sinovac, BBIBP-CorV by Sinopharm, and BBV125 COVAXIN by Bharat Biotech) or just the S protein, which can be delivered in the form of messenger RNA (e.g., BNT162b2 by Pfizer-BioNTech and mRNA-1273 by Moderna), adenoviral vector (e.g., ChAdOx1-S by Oxford/AstraZeneca, Ad26.COV2.S by Janssen, and Gam-COVID-Vac by Gamaleya), or recombinant protein (e.g., NVX-CoV2373 by Novavax). According to the results from phase III clinical trials, the protection against the wild-type SARS-CoV-2-induced disease ranges from 94 to 95% of two doses of mRNA-based vaccines [124,125] to 67–74% of one or two doses of adenoviral vector-based vaccines [126,127]. The efficacy of the other vaccines lies in between: it was about 89% for the protein-based vaccine [128] and 78–83% for those using inactivated viral particles [129,130].

The primary output of vaccine effectiveness is the production of nAbs that can potentially prevent the infection, providing a sterilizing immunity. Therefore, the presence and abundance of S-specific nAbs provide the strongest correlate of protection from subsequent infection with the same viral strain and disease development [131,132]. However, studies performed in relevant animal models demonstrate that antigen-specific T cells induced by vaccination contribute to the efficacy of vaccines [133,134]. They can extend the duration of vaccine-induced protective immunity after circulating antibodies start waning and broaden the protection against antibody-escaping virus variants.

S-specific CD4^+^ T cells are found in nearly all human subjects receiving two doses of mRNA COVID-19 vaccines, and memory CD4^+^ T cells are maintained up to 6 months after the second dose [135,136,137,138]. The frequencies of S-specific CD4^+^ memory T cells elicited by the two doses of mRNA COVID-19 vaccines and their distribution among the T_CM_, T_EM_, and T_EMRA_ subsets are comparable with those induced by natural infection, and the patterns of contraction and estimated half-lives are also similar [137,138,139,140]. Interestingly, vaccine-induced CD4^+^ memory T cells include a fraction of T_SCM_ cells that may support establishing long-term memory [139]. Looking at the effector function of S-specific CD4^+^ T cells, they are mainly cT_FH_ cells and T_H_1 cells producing IFN-γ, TNF-α, and IL-2 [141]. As for the natural infection, the frequency of vaccine-induced cT_FH_ cells positively correlates with nAb titers [135,142]. Notably, the frequency of vaccine-induced cT_FH_ and T_H_1 cells after the first immunization correlate with the abundance of nAbs and frequency of S-specific CD8^+^ T cells following the second dose of the vaccine, highlighting the role of rapidly stimulated CD4^+^ T cells in coordinating the immune response to the second vaccine dose, especially in individuals who did not experience previous SARS-CoV-2 infection [140]. Moreover, although the frequency of S-specific cT_FH_ cells peaks one week after the second immunization, S-specific T_FH_ cells in lymph nodes persist at least for 6 months [143], and their impairment is strictly associated with compromised germinal center reactions and nAbs production, as shown in immunocompromised individuals undergoing kidney transplantation [144]. S-specific polyfunctional T_H_1 cells and cT_FH_ cells are comparably induced by adenoviral vector-based and recombinant protein-based vaccines [138,145], although it is difficult to assess a quantitative side-by-side comparison, also due to different immunization schedules [138].

S-specific CD8^+^ T cells are detected in about 70–90% of subjects who received two doses of mRNA vaccines. However, memory CD8^+^ T cells are maintained only in 40–65% of people six months after the second dose [135,136,137,138]. Moreover, the magnitude of CD8^+^ T-cell memory is lower than CD4^+^ memory T cells [146,147]. Nonetheless, vaccine-induced CD8^+^ T cells are polyfunctional; they are able to produce different effector molecules, such as IFN-γ, TNF-α, and granzyme B; and, like S-specific CD4^+^ T cells, their distribution among T-cell subsets mirrored the one observed in a natural infection, with a prevalence of T_EM_ and T_EMRA_ cells [138,139,141,147]. Notably, a vaccine based on adjuvanted recombinant S protein elicits a very low frequency of S-specific CD8^+^ memory T cells compared with mRNA vaccines [138], with minimal effects on the protection from the disease measured during clinical trials, indicating a superior role for CD4^+^ T cells in establishing the vaccine-induced protective immunity.

The knowledge about generating S-specific T_RM_ cells upon vaccination is still inadequate. However, one study reported the absence of vaccine-induced S-specific T cells in the BAL fluid of vaccinated subjects despite their detection in peripheral blood [148]. Similarly, a second study showed a very limited induction of S-specific T_RM_ cells in lung biopsies following mRNA vaccination compared to infection [149].

These data demonstrate that COVID-19 vaccines trigger the development of S-specific CD4^+^ memory T cells and, often, CD8^+^ memory T cells. In particular, S-specific CD4^+^ T cells have a pivotal role in orchestrating the humoral and cellular responses to the vaccine. Moreover, vaccine-induced S-specific CD4^+^ and CD8^+^ memory T cells have a functional phenotype similar to that observed in response to natural infection and are predicted to be long-lived. On the contrary, COVID-19 vaccines seem unable to elicit the differentiation of S-specific T_RM_ cells (Figure 4).

## 7. T-Cell Responses to SARS-CoV-2 Variants and Hybrid Immunity

Despite having evolved an RNA proofreading mechanism acting during replication [150], SARS-CoV-2 has accumulated many mutations over time. Most of these mutations do not modify the amino acid sequence of viral proteins or do not provide any evolutionary benefit. However, some mutations that provide survival advantages and improved viral “fitness” have spread worldwide and are defined as variants of concern (VoCs). The major VoCs include Alpha (B.1.17 lineage), Beta (B.1.351), Delta (B.1.617), and Omicron (B.1.1.529). In particular, Delta and Omicron VoCs are characterized by significantly higher transmissibility and infectivity. The presence of multiple mutations in the S protein, especially in the RBD domain, allows these VoCs to escape, at least in part, the immune protection mediated by nAbs generated in response to previous infection and vaccination [151,152]. Moreover, Omicron can escape the neutralizing activity of most, but not all, of the therapeutic monoclonal antibodies currently available for clinical use [153].

The ability of VoCs to escape the control of vaccine-induced nAbs has raised concerns. However, several pieces of evidence show that vaccine-induced T cells can recognize all the SARS-CoV-2 variants of concern, including Omicron, largely preserving the protection against severe COVID-19 [136,137,141,154,155]. This is possible because S-specific CD4^+^ and CD8^+^ T cells recognize a median of 10–11 Spike epitopes in each person, and the great majority of these epitopes are conserved across VoCs [136].

The emergence of antibody-escaping VoCs and the drop in nAb titers 6–8 months after vaccination induced many countries to implement the third dose of vaccine (booster) at about 6 months from the second dose, especially to protect fragile subjects. The booster dose of mRNA vaccine promptly restores S-specific antibody titers and elicits potent neutralization across different VoCs, including Omicron, at least for three months after vaccination [156,157,158]. The enhanced functionality of the induced nAbs derives from the reexpansion of preexisting memory B-cell clones and the stimulation of new B-cell responses with increased potency and breadth [159]. Evidence from other vaccines with a three-dose schedule indicates that durable Ab responses are triggered after the third dose [146], but evaluating the durability of the protective immunity induced by the COVID-19 vaccination booster will require more time and additional studies. S-specific T cells are also rapidly restimulated after the booster vaccine [160], even in a good proportion (>50%) of patients with compromised immune responses secondary to different diseases or therapies [161,162]. However, data about T cells are largely limited to the measurement of IFN-γ production, and additional information on their functional phenotype is still missing.

The vaccination of people who had been previously infected by SARS-CoV-2 and the infection of vaccinated people by SARS-CoV-2 VoCs have provided additional insights on T-cell responses to SARS-CoV-2 following repeated exposures as well as on hybrid immunity, intended as a combined immune response to natural infection and a vaccine. S- and RBD-specific memory B-cell frequencies substantially increase and have higher somatic hypermutation and affinity maturation in hybrid immunity than after vaccination [137,163,164]. Consequently, nAbs titers and breadth of neutralization of VoCs are significantly improved in hybrid immune responses compared with the vaccination or the infection alone [146]. On the contrary, there are only modest differences in the frequency of circulating S-specific CD4^+^ and CD8^+^ T cells and their IFN-γ production capacity between hybrid immunity and vaccination [140,165,166,167]. However, data on the quality of T-cell responses elicited by hybrid immunity are still scattered. A study reported that although the order in the type of exposure (infection or vaccination first) affects the distribution between S- and non-S-specific T-cell responses, there is no evidence of major alterations in the TCR repertoire of epitope-specific CD8^+^ T cells upon repeated exposure [168]. The same study showed that repeated stimulations lead to a shift in SARS-CoV-2-specific CD8^+^ memory T cells toward T_EMRA_ cells, but these cells are not exhausted [168]. Another report showed, instead, that vaccination can induce a repertoire of S-specific CD4^+^ T-cell clones that substantially diverges from the one previously activated by the infection [169], thus further broadening the antigen-specific T-cell response against the virus. Similarly, SARS-CoV-2 breakthrough infections have been reported to enhance the magnitude, breadth, and repertoire of T-cell responses [170]. Another possible advantage of hybrid immunity compared to vaccination alone is the generation of T_RM_ cells [171] observed after SARS-CoV-2 infection but not upon vaccination (Figure 4). However, further investigation of vaccine-induced T-cell tissue immunity is required to clarify this topic.

Together, these data indicate that memory T cells induced by infection and vaccination can protect against SARS-CoV-2 antibody-escaping variants thanks to a polyclonal response that can recognize multiple conserved epitopes.

## 8. Conclusions and Future Perspectives

The optimal protection against SARS-CoV-2 infection requires all the components of the innate and adaptive immune systems to function coordinately. When this happens, SARS-CoV-2 infection is resolved in a few days without or with mild symptoms. On the contrary, an impairment or delay in activating one of these components, due to virus-dependent or host-dependent factors, results in an uncoordinated response that can lead to severe disease. Upon infection, the adaptive immune system develops a memory of SARS-CoV-2 that generates enhanced immune responses and protection in case of reexposure to the same virus. COVID-19 vaccines are designed to mimic the SARS-CoV-2 infection without disease to train the adaptive immune system to develop an immune memory that protects us when we are naturally exposed to the virus.

The immunological memory is mediated by four different arms of the adaptive immune system: antibodies, memory B cells, and memory CD4^+^ and CD8^+^ T cells. Antibodies are the only component that can prevent infection. Their continuous production is sustained by long-lived plasma cells, but it is clear that circulating SARS-CoV-2 nAbs levels decline some months after infection or vaccination. Moreover, the emergence of viral variants has made it evident that reaching herd immunity and completely blocking the circulation of the virus are targets currently not achievable. Nonetheless, memory T cells can efficiently protect us from severe disease and, in part, from symptomatic infection. SARS-CoV-2-specific memory T cells can recognize several conserved viral epitopes and protect against SARS-CoV-2 variants that breach the antibody barrier. Therefore, the efficacy of vaccines in establishing protective immunity should be routinely evaluated not only based on the capacity to elicit the production of nAbs, as currently performed, but also on the ability to generate antigen-specific memory T cells. However, T cells can recognize antigens only in the context of MHC presentation, making the ex vivo identification and characterization of antigen-specific T cells difficult and hindering the development of high-throughput screening platforms. This aspect will require technical and technological improvements, which will be critical not only for clinical applications but also for the fundamental investigation aimed at increasing our knowledge of the mechanisms guiding the development of human memory T cells.

Approved COVID-19 vaccines have been developed extremely quickly thanks to the knowledge acquired in the last decades and have proven to be highly effective in protecting against severe COVID-19 but not from reinfection, also due to the emergence of viral variants. Currently, pharmaceutical companies are updating COVID-19 vaccines with the Spike sequence of the emerging SARS-CoV-2 variants, but they are evolving too fast to keep pace [172]. The development of a pan-coronavirus vaccine would overcome this issue and have a great epidemiologic and economic value [173]: about fifteen of these vaccine candidates are in development [174,175,176]. They have different degrees of target breadth, ranging from SARS-CoV-2 variants to the whole coronavirus genus, but they are still far from reaching clinical use [175,176].

In the meantime, there is room for improving approved vaccines. First, the observation that circulating nAbs titers decline a few months after vaccination suggests that current vaccines cannot efficiently trigger the differentiation of long-lived plasma cells [177]. Second, approved vaccines, different from natural infection, fail to generate resident memory T and B cells and mucosal Abs. T_RM_ cells can provide a rapid reaction of the immune system when the virus infects peripheral tissues, such as the nasopharyngeal mucosa and the lungs, thus preventing viral spreading from the site of entry and symptomatic disease. The differentiation of vaccine-induced T_RM_ cells may depend on the route of immunization [134,178,179], the kind of antigen [180], or the use of specific adjuvants [181,182]. The “prime-and-pull” vaccination strategy has proven to generate both systemic and local memory T cells [183] and was effective in reducing Herpes simplex virus type-2 recurrent infections in preclinical studies [184]. In this setting, a conventional parenteral vaccination that elicits a systemic T-cell response (prime) is followed by the local application of the antigen or a chemoattractant (pull) to establish a pool of T_RM_ cells [183,185]. Interestingly, multiple parenteral immunizations can also induce the differentiation of T_RM_ cells [186], although with lower efficiency compared with the local antigen administration [187], suggesting that the third or the fourth dose of current vaccines may generate S-specific T_RM_ cells to a certain extent. A “prime-and-pull” strategy could be recapitulated for COVID-19 vaccines by combining the current schedules of intramuscular vaccination with a boost of a locally administered vaccine, such as through the nasal route. Intranasal COVID-19 vaccines are under development [188] and have shown promising results in animal models [189,190,191]. Upon completion of clinical trials, they may represent an additional weapon in the fight against COVID-19.

Another element that may be improved to ameliorate vaccine efficacy is the adjuvant. Adjuvants are vaccine components that stimulate the immune system to enhance the magnitude, breadth, and durability of the vaccine-induced immune response when these signals are not provided by the antigen, namely, in all vaccines except for those made of live attenuated pathogens [192]. At the moment there are very few vaccine adjuvants approved for clinical use, and while they have been extensively tested for their capacity to enhance Abs production, the knowledge of their effect on memory T-cell differentiation is surprisingly scarce [192]. Developing vaccine adjuvants that can induce the generation of long-lived memory T cells with the proper polarization and homing capacity will certainly improve the efficacy of vaccines.

To conclude, the COVID-19 pandemic has greatly challenged humanity with huge social and economic costs and the loss of millions of lives. At the same time, it has boosted unprecedented cooperation in the scientific community and has led to the rapid development of therapeutic and preventive strategies to tackle the emergency. Extraordinary progress has been quickly made in disentangling and understanding the immune response to SARS-CoV-2 infection and vaccination, and this knowledge will be precious in guiding the development of improved vaccines against SARS-CoV-2 and other diseases.

## Figures and Tables

**Figure 1 vaccines-12-01126-f001:**
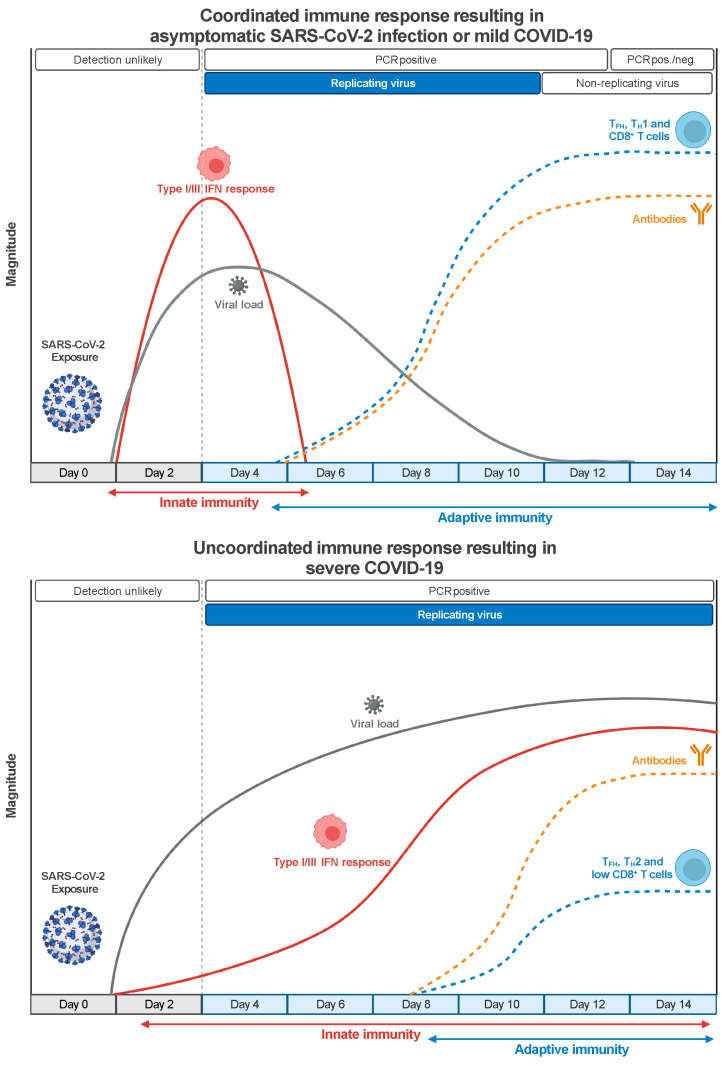
**Coordinated innate and adaptive immune responses to SARS-CoV-2 infection correlate with better clinical outcomes.** Protective immunity to SARS-CoV-2 infection requires the prompt stimulation of the innate immune system that, through the activation of type I/III IFN pathways, limits viral replication and triggers the timely induction of adaptive immune responses. In turn, SARS-CoV-2-specific T_H_1 and T_FH_ CD4^+^ T cells optimize the effector function of cytotoxic CD8^+^ T cells and the production of potent neutralizing Abs, leading to the resolution of the infection in about two weeks. On the contrary, the defective activation of type I/III IFN responses results in unrestrained viral replication, delayed activation of the adaptive immune response, and sustained late production of proinflammatory cytokines. The impaired induction of SARS-CoV-2-specific cytotoxic CD8^+^ T cells and the wrong polarization of CD4^+^ T cells results in a reduced capacity to control the infection, while excessive inflammation causes extensive tissue damage.

**Figure 2 vaccines-12-01126-f002:**
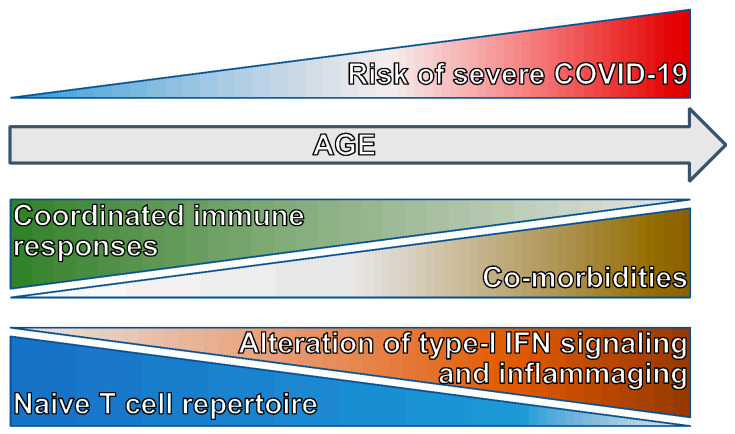
**Correlates of increased severe COVID-19 risk with aging.** The risk of developing severe COVID-19 is significantly higher in the elderly. Several factors contribute to the increased risk, including an elevated incidence of comorbidities, a reduced repertoire of naïve T cells, alterations in type I/III signaling, and a higher basal inflammatory level. These factors lead to the uncoordinated innate and adaptive immune responses underlying severe COVID-19.

**Figure 3 vaccines-12-01126-f003:**
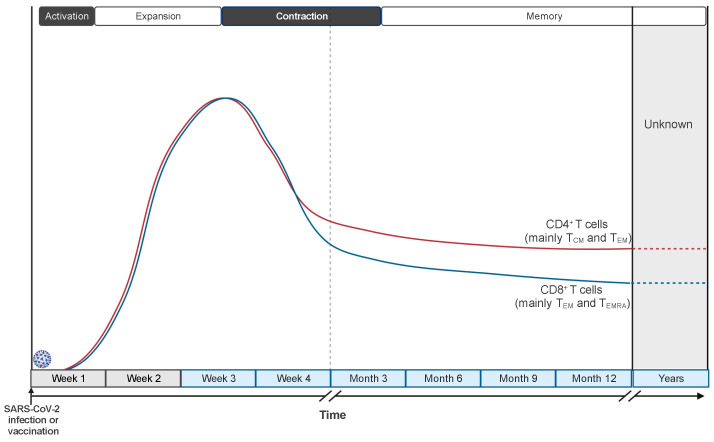
**Kinetics of CD4^+^ and CD8^+^ T-cell responses to SARS-CoV-2 infection and vaccination.** SARS-CoV-2-specific T cells are detected as early as 3–5 days after symptoms onset and expand in the following 2–3 weeks. Then, their frequencies start to decline with kinetics that, in the first two months, are slightly faster for CD8^+^ T cells than CD4^+^ T cells. SARS-CoV-2-specific memory CD4^+^ and CD8^+^ T cells are detectable up to 12–15 months after infection or vaccination and predicted to be long-lived.

**Figure 4 vaccines-12-01126-f004:**
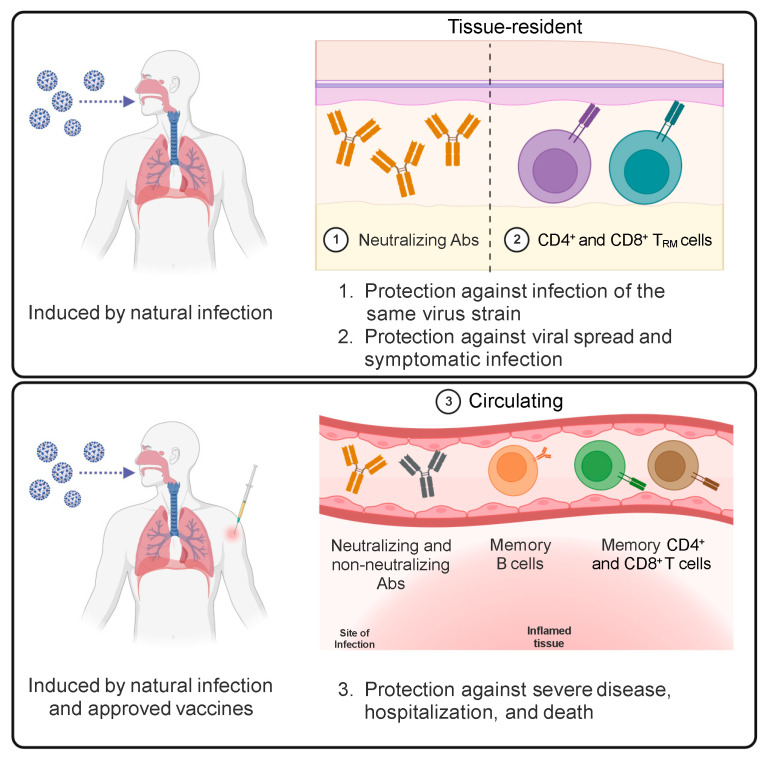
**Protective immunological memory induced by SARS-CoV-2 infection and vaccination.** SARS-CoV-2 infection and COVID-19 vaccines elicit the production of circulating neutralizing and non-neutralizing Abs and induce the differentiation of memory B cells, CD4^+^ T cells, and CD8^+^ T cells that can significantly protect against severe COVID-19, hospitalization, and death. The natural infection also stimulates the differentiation of mucosal neutralizing Abs and CD4^+^ and CD8^+^ T_RM_ cells that provide local protection at the site of viral entry and avoid virus spread. Neutralizing Abs are the only component of the adaptive immune system that can prevent infection and generate sterilizing immunity.

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
