# Peer review of "T-Cell Immune Responses to SARS-CoV-2 Infection and Vaccination"

_vaccines, 2024, doi:10.3390/vaccines12101126_

Round 1
Reviewer 1 Report
Comments and Suggestions for Authors
This review article on T cell immune responses to SARS-CoV-2 infection and vaccination is a well-written document that allows the reader to update himself on the main aspects of the immune response to SARS-CoV-2, both to the vaccine and to natural infection. It is accompanied by 4 figures written by the author, which illustrate very well the different moments of the immune response and its relationship with the clinical outcome of patients, as well as a figure where he clearly explains the worse prognosis observed in older people when faced with COVID-19.
It is pleasant and clear to read.
I only have one observation on line 123 of page 4, when the effector function of the complement is mentioned: It is NOT clear, they should explain better, since the FC receptors are on the surface of some immune cells, while the activation of the complement by the classical pathway occurs by the binding of the Fc region of IgG or IgM to the C1 fraction of the complement.

Author Response
I thank the Editors and Reviewers for appreciating my effort in providing a comprehensive overview of the T cell-mediated immune response to SARS-CoV-2 infection and COVID-19 vaccine and suggesting how to improve the manuscript. The referees have made positive and beneficial comments, to which I provide a point-by-point response below.
Comment #1:
I only have one observation on line 123 of page 4, when the effector function of the complement is mentioned: It is NOT clear, they should explain better, since the FC receptors are on the surface of some immune cells, while the activation of the complement by the classical pathway occurs by the binding of the Fc region of IgG or IgM to the C1 fraction of the complement.
Reply to comment #1:
Thank you for highlighting the ambiguity of this sentence, which resulted from an excess of synthesis. Here, I meant to briefly describe two separate mechanisms by which antibodies can promote the immune effector function: on the one hand, the binding to Fc receptors on immune cells, such as NK cells, to induce antibody-dependent cellular cytotoxicity; on the other hand, the activation of the complement resulting in the lysis of target cells.
The revised sentence will read as follows.
“Antibodies recognize viral epitopes in their native conformation and can block the binding of the viral Spike protein to the ACE2 receptor on human cells, thus preventing infection. Moreover, antibodies can promote the killing of virus-infected cells, either by binding to Fc receptors on NK cells and inducing the antibody-dependent cellular cytotoxicity or by activating the classical pathway of the complement, which leads to the lysis of the infected cells and facilitates their clearance by phagocytes.”
Reviewer 2 Report
Comments and Suggestions for Authors
The manuscript by Samuele Notarbartolo provides a comprehensive review of the immune response to SARS-CoV-2 infection and vaccination. While it acknowledges the roles of both innate and adaptive immunity in controlling SARS-CoV-2, the paper focuses primarily on T cell responses. It covers key topics such as alterations in T cell populations in COVID-19 patients, SARS-CoV-2 T cell antigens and immunodominant epitopes, SARS-CoV-2-specific T cell responses to infection and vaccination, and T cell responses to SARS-CoV-2 variants and hybrid immunity.
The manuscript is well-written, providing robust scientific insights that significantly contribute to our understanding of the T cell response to SARS-CoV-2 infection and vaccination. I have only two minor questions, which do not affect the overall high impact of the paper:
1) Organ transplant patients receive daily immunosuppressive treatment. SARS-CoV-2 infection and COVID-19 vaccination in these patients could offer valuable insights into how immune suppression affects T cell and B cell responses. Only one paper is cited regarding S-specific Tfh cells in kidney transplant recipients. Could more literature on SARS-CoV-2 infection and COVID-19 vaccination in transplant patients be reviewed?
2.) Is T cell exhaustion linked to severe COVID-19?
Author Response
I thank the Editors and Reviewers for appreciating my effort in providing a comprehensive overview of the T cell-mediated immune response to SARS-CoV-2 infection and COVID-19 vaccine and suggesting how to improve the manuscript. The referees have made positive and beneficial comments, to which I provide a point-by-point response below.
Comment #1:
Organ transplant patients receive daily immunosuppressive treatment. SARS-CoV-2 infection and COVID-19 vaccination in these patients could offer valuable insights into how immune suppression affects T cell and B cell responses. Only one paper is cited regarding S-specific Tfh cells in kidney transplant recipients. Could more literature on SARS-CoV-2 infection and COVID-19 vaccination in transplant patients be reviewed?
Response to comment #1:
Thank you for mentioning this topic. Indeed, studies on transplanted patients undergoing intensive immunosuppressive treatment further corroborate the key role of the immune system in tackling SARS-CoV-2 infection. However, it is difficult to understand from these studies the relative contribution of the different branches of the immune system in clearing the virus.
I have added a small paragraph discussing the topic in lines 146-160 of the revised manuscript.
Comment #2:
Is T cell exhaustion linked to severe COVID-19?
Response to comment #2:
Thank you for raising the issue. The scientific community still debates a possible association between T cell exhaustion and severe COVID-19 (Rha MS, Shin EC. 2021. doi: 10.1038/s41423-021-00750-4). In my opinion, the concept of T cell exhaustion does not fit well in the context of an acute viral infection, because it normally results from the chronic stimulation of T cells. Much confusion is generated by the fact that, in many papers, researchers tend to refer to T cells expressing inhibitory receptors and other exhaustion-associated molecules as exhausted T cells. However, inhibitory receptors are usually upregulated in recently activated T cells to restrain their effector function on time and avoid excessive immune responses. Instead, T cell exhaustion is a permanent (unless treated) dysfunctional state requiring epigenetic changes and metabolic reprogramming and marked by an elevated and persistent expression of inhibitory receptors.
I have added this point in lines 370-380 of the revised manuscript. I had already mentioned in the original manuscript that repeated vaccination and hybrid immunity do not induce T cell exhaustion (lines 567-569 of the revised manuscript).